# Parkour and Intrinsic Motivation: An Exploratory Multimethod Analysis of Self-Determination Theory in an Emerging Sport

**DOI:** 10.3390/ijerph22111632

**Published:** 2025-10-27

**Authors:** Jacob Carson, Samantha Hurst, James F. Sallis, Sarah E. Linke, Eric B. Hekler, Katherina Nardo, Britta Larsen

**Affiliations:** Herbert Wertheim School of Public Health & Human Longevity Science, University of California, La Jolla, San Diego, CA 92093, USA; shurst@health.ucsd.edu (S.H.); jsallis@health.ucsd.edu (J.F.S.); slinke@health.ucsd.edu (S.E.L.); ehekler@health.ucsd.edu (E.B.H.); knardo@lmri.net (K.N.); blarsen@health.ucsd.edu (B.L.)

**Keywords:** multimethod, physical activity, lifestyle sports, qualitative, adolescents

## Abstract

Self Determination Theory posits that individuals may be more likely to initiate and maintain behaviors tied to intrinsic (vs. extrinsic) motivations and may provide a useful framework for understanding youth participation in novel sports. Using the Intrinsic Motivation Inventory (IMI) and Patient-Centered Assessment and Counseling for Exercise Plus Nutrition (PACE+) surveys, motivation and physical activity habits were explored in 27 children/adolescents (ages 7–16) enrolled in Parkour, an individual, non-competitive youth sport. Fifteen Parkour participants were also interviewed to gain an understanding of their motivations for participating. Study participants had high median IMI subscale scores related to interest/enjoyment (6.71/7) and perceived choice (6.40/7) compared to the whole scale. Similarly median sub-scale Pros and Self-Efficacy scores for physical activity from the PACE+ were high (4.25/5 and 3.91/5, respectively). The themes of autonomy and enjoyment were consistently reported in the qualitative interviews, expanding on the quantitative results. Other themes included appreciation for camaraderie, creativity, and a drive for improvement. These results provide early evidence that Parkour, and similar lifestyle sports, may be tied more to intrinsic than extrinsic motivations and could have potential for adoption and maintenance by youth with low motivation to engage in physical activity to promote healthy behaviors.

## 1. Introduction

Current physical activity (PA) guidelines recommend that children (ages 6–17) should engage in 60 min of daily moderate-to-vigorous PA [1]. PA for children and adolescents increases cardiorespiratory and muscular fitness, improves cardiometabolic and bone health, and lowers adiposity [2]. According to the CDC, only 24% of US children meet PA guidelines, while obesity prevalence in this age group is as high as 18.5% [3,4]. Establishing healthy PA habits at a young age is an important strategy for creating population-level health change [5]. PA declines with age and this decline in leisure-time PA continues from adolescence to adulthood [6,7], reinforcing the importance of PA interventions among children and adolescents across all stages of development.

Understanding the factors that influence children’s motivations to participate in PA can inform appropriate interventions. Self-Determination Theory (SDT), a psychological framework that focuses on the relationship between intrinsic and extrinsic motivators for promoting behavior, may provide a useful framework to understand motivation for PA in children. Meta-analyses show that intrinsic motivation (inherent interest and enjoyment), as opposed to extrinsic motivation (motivated by the outcome or a reward), is more strongly associated with exercise adoption and long-term maintenance in both adolescents and adults [8,9,10]. One of the six mini-theories of SDT is the Basic Psychological Needs Theory (BPNT) specifically posits that three factors make up the basic psychological needs (BPNs): Autonomy (perception of control), Competence (perception of capability), and Relatedness (perception of social connection) [11]. According to BPNT, having these psychological satisfied results in more internally regulated types of motivation, which have been consistently associated with higher levels of PA [9,10,12]. On the contrary, frustration of these BPNs can result in demotivation [11].

Previous experimental approaches have supported the association between SDT constructs and PA, particularly intrinsic motivation and the novelty of the activity, however methodological inconsistencies and heterogeneity in findings have limited broader conclusions about specific activities [12,13,14,15]. Specifically, autonomous forms of motivation predict PA in children more positively and strongly than their controlled (non-autonomous) motivation counterparts [12]. More recent research has explored the relative importance of the BPNT constructs, highlighting the need to address competence to promote autonomous motivation [13]. These findings are echoed in the qualitative literature as well, finding that physical education aligned with these theories may improve student outcomes, specifically facilitated through peer relationships [16]. Previous research from Spain and Argentina have identified that satisfaction of BPNs among adolescents resulted in increased motivation to engage in leisure-time physical activity, and that the satisfaction of these needs differed between structured and unstructured activities [17,18].

Additionally, sport discontinuation among children appears to be consistent with the SDT framework. Previous research has found that sport participation significantly decreases with age [19,20]. The most commonly cited reason for discontinuing participation in a sport is lack of fun, followed by stress, negative team dynamics, and competition [8]. More recent research has explicitly found the explanatory power of BPNT for explaining sport continuation [21]. A systematic review from 2015 also highlights factors related to competence autonomy and relatedness as reasons for sport discontinuation among youth [22]. In the case of competitive team sports, extrinsic motivation may remain high for children who excel and move towards specialization, while those who do not excel and/or do not enjoy competition may play less and be less likely to continue participating [23]. Despite this theoretical alignment, research on how specific sports/activities (versus leisure-activity or sport more generally) may support or impede autonomous motivation has been limited.

Motivations may differ for adoption of and sustained engagement in non-competitive, individual level “lifestyle” sports and could better align with the BPNs [24,25]. In Parkour, for example, individuals practice movement skills based on running, jumping, and climbing with the goal of improving their strength, balance, and adaptability. Parkour also encompasses movements from other disciplines including gymnastics and martial arts, allowing for more creative expression through acrobatics [26]. Parkour, a traditionally non-competitive discipline, was founded in the late 1980s in France, based off a military practice of using obstacle courses to promote functional fitness and self-improvement. In the years since its development, Parkour has evolved to have both leisure-time activity and sport framings for different groups. The non-competitive, creative, and more generalized fitness approach provided through Parkour more generally may support intrinsic motivation and therefore sustained engagement in PA for youth. Despite growing popularity through social media and talks of potential Olympic inclusion [27], Parkour is still niche and there has been limited academic research on Parkour for its potential for PA interventions or programs with youth. Parkour may provide a unique perspective and approach to youth engagement and sport that warrants further study through the lens of established theory [28].

The aim of the current study is to explore children’s intrinsic and extrinsic motivations for participating in Parkour. Specifically, we used an exploratory multimethod approach to examine the motivations, behaviors, and attitudes of youth who participated in Parkour as a leisure-time activity. We then conducted semi-structured individual interviews with youth participating in Parkour to expand upon the findings and better understand motivation for initiating and maintaining participation in Parkour through the lens of SDT and BPNT more specifically.

## 2. Materials and Methods

This study utilized both quantitative and qualitative methodologies, following an explanatory sequential mixed methods approach with the initial collection of quantitative survey data followed by qualitative interviews to provide a broader framework of understanding triangulate the findings [29]. Quantitative data were cross-sectional and collected via self-report surveys. Qualitative interview data collection took place via semi-structured individual interviews. All study procedures were approved by the university institutional review board, and parental consent/participant assent was given prior to participation.

### 2.1. Setting

Data were collected from November 2019 to April 2020. The primary site for survey data collection was APEX School of Movement San Diego; a Parkour facility located in central San Diego. Semi-structured individual interviews were conducted remotely via Zoom software (version 4.6.9).

### 2.2. Eligibility

To be eligible for this study, participants had to be between 7 and 17 years of age, be able to speak, write and read English, and have an active membership at APEX San Diego. Potential participants were excluded if they had a medical condition that prevented them from participating in any PA, as those with prohibitive medical conditions may have significantly different perspectives or habits related to motivation and exercise.

### 2.3. Quantitative Data Collection

#### 2.3.1. Recruitment

Twenty-seven participants were initially recruited in-person at the Parkour facility through flyers and word of mouth. Following Shelter in Place orders in California due to the COVID-19 pandemic in March 2020, all research activities were moved online, included recruitment for qualitative interviews.

#### 2.3.2. Variables and Measures

All study participants provided data on their age and sex. Two surveys were used to measure motivation and perceptions of PA: The Intrinsic Motivation Inventory (IMI) and the Patient-Centered Assessment and Counseling for Exercise Plus Nutrition (PACE+) for Adolescents surveys (see below). Facility visit data accessed from the Parkour gym database including the number of unique visits (excludes back-to-back classes or open gym use) to the gym and the number of months they have been attending the gym to assess if time involved in Parkour or frequency of participation had any significant association with the variables of interest. Surveys were administered by the PI under the supervision of the participants’ parents. The participants were tasked with completing the survey independently while on-site so that any clarifying questions could be answered by the PI.

The Intrinsic Motivation Inventory (IMI) is a multidimensional measure of task-related experience based in SDT. This was chosen as a direct measure for intrinsic motivation. The standard 22-item version assesses four subscales: interest/enjoyment, made up of seven items, and perceived competence, perceived choice, and pressure/tension, each made up of five items [30]. Each item is scored on a 7-point Likert scale, with a one representing the lowest score for the item and a seven representing the highest. The tool has shown high Cronbach alphas for internal consistency for the four subscales: interest/enjoyment (α = 0.78), perceived competence (α = 0.80), effort (α = 0.84), and pressure/tension (α = 0.68) [31]. This scale and its subscales have been used in multiple adolescent populations since validation [32,33,34]. The four subscale scores of each participant were calculated and used in data analyses. As a reference of average scores, previous use of this scale in youth resulted in mean subscale scores of 4.7 for interest/enjoyment, 5.7 for perceived competence, 3.9 for perceived choice, and 2.4 for pressure/tension [35].

The Physician-Based Assessment and Counseling for Exercise (PACE+) Adolescent survey is a validated measure of adolescents’ PA, sedentary behavior, and diet designed for use in primary care settings [36]. This measure was used to estimate self-reported PA as well as psychosocial variables such as self-efficacy and peer influences on PA. The entire survey has seven sections comprising a total of 47 questions, primarily on a Likert-scale. This measure has been validated in adolescents against the criteria of PA stages of change and objective measures of PA [36,37]. Stage of Change, from the Transtheoretical Model of Change, represents the 5 stages that individuals move through in adopting health behaviors (Precontemplation, Contemplation, Preparation, Action, Maintenance), and in this survey serves to contextualize where adolescents are in achieving healthy activity habits [36]. Similarly, Change Strategies represents the self-reported strategies that participants have to move along that continuum towards Maintenance. In addition to Stage of Change (3 items), the survey provides seven subscale scores (Likert scale ranges from 1 to 5) That were used in the final analyses: Change Strategies (15 items, α = 0.88), Pros (5 items, α = 0.81), Cons (5 items, α = 0.53), Self-Efficacy (six items, α = 0.76), Family Influences (four items, α = 0.79), Friend Influences (six items, α = 0.60), and Environmental Influences (four items, α = 0.42). From a previous validation study, the mean subscale scores were as follows: 3.2 for Change Strategies, 3.4 for Self-efficacy, 4.0 for Pros, 2.1 for Cons, 2.8 for Family Influences, 3.2 for Peer Influences, and 3.6 for Environmental Factors [37].

#### 2.3.3. Analyses

Survey data were analyzed for descriptive results. Given the nature of the variables, frequencies were calculated for Sex and Stage of Change, while univariate statistics were calculated for all other variables. Given the small n, we computed both means and medians for participant characteristics. Participants with missing data were not included in calculations for the corresponding subscales. The data analysis for this paper was generated using SAS Studio Version 3.81, (SAS Institute Inc., Cary, NC, USA).

### 2.4. Qualitative Data Collection

#### 2.4.1. Recruitment

All participants from the initial survey were contacted via email and asked about their willingness to participate in a follow-up individual interview. A total of 15 individuals who participated in the initial survey agreed to the qualitative interview, over half of our quantitative sample.

#### 2.4.2. Interview Structure

Interviews were conducted remotely using Zoom, a teleconferencing platform [38]. All interviews were conducted during the months of March and April in 2020 by a female research staff member (K.N.) with prior interview experience who had no personal relationship with the students. Interviews lasted approximately 30 min on average. Parental consent and child assent were given to collect the audio recordings of the interviews prior to the start of the interview, and no field notes were recorded. The interview guide was first piloted with two participants, after which three sub-questions were added to clarify the subject domains in the interview guide for the remaining 13 participants (see the Interview Guide in the Appendix A). The recruitment goal of 15–20 participants for the qualitative portion of the study was determined based on the proportion of total study participants (~50%), participant availability, and thematic saturation based on our theoretical framing [39].

Questions included in the interview guide were designed to be easily understood for the entire age group (as young as seven) and covered such topics as how they became involved in Parkour, what they like about it compared to other sports, their goals for the sport, and their perceptions of competition to complement the quantitative data collection. Interview guide questions were developed with the assistance of an experienced qualitative researcher on the team. The goal of the interviews was to capture a broad understanding of how they felt about Parkour through the framing of SDT and BPNT more specifically. The primary sections of the interview guide included: autonomy and perceptions (how they feel about parkour and their choice in participation), goals and motivations (what they strive to get out of parkour), and competition (how they feel about competition and how that impacts their activity choices). The interview length was purposively kept short (~15–20 min) to minimize the burden on the children, and question topics were more directed to be age-appropriate (i.e., “What kinds of things make you excited to come to Parkour classes?”).

Interview recordings were transcribed by the PI using Otter.ai, a secure online platform that automatically performs a rough transcription [40]. The interview transcripts were reviewed against the audio recording to correct any inaccuracies during the transcription process.

#### 2.4.3. Analyses

A directed content analysis approach was used in line with the aims of the study [41]. Complete transcriptions were uploaded onto Dedoose, a web-based qualitative analysis platform [42]. The coding schema for the qualitative data were developed a priori from the interview guide questions, with a specific focus on SDT constructs of interest including autonomy, relatedness, and competence. Additional descriptive codes were added as needed to label novel or emergent content not outlined by the interviews, respectively. All excerpts selected during this process were summarized for consistent language and ease of understanding before moving on to secondary coding, which was performed using both focused coding (inclusively categorize coded data on thematic similarity) and axial coding (relating categories and subcategories of codes into patterns for similarity). The resulting visual maps and direct quotations from study participants resulted in three major overlapping themes aligned with the specific aims of the study. Coding was performed primarily by the PI and reviewed with a qualitative research specialist to maintain a high level of rigor. Any disagreements in the application of primary and secondary coding labels were resolved through discussion and consensus between the PI and qualitative research specialist. Due to the study design, participant checking was not performed, and transcripts were not returned to the participants. Qualitative themes and findings were then compared with quantitative findings to contextualize and enhance interpretation across our theoretical framework.

## 3. Results

### 3.1. Quantitative Results

A total of 27 participants agreed to participate in first stage of the study. The mean age was 11.07 years (SD = 2.66), with 25 participants being male and only two being female (Table 1). Eleven participants (40.74%) were classified in the “Maintenance” stage for PA, while seven and nine participants were classified as Contemplative and Pre-Contemplative, respectively. There was a wide range of Parkour experience in the study sample represented by the large ranges in both Months of Attendance and Number of Gym Visits.

Relative to the aforementioned subscale means from previous use of the IMI in youth, this sample scored highly in interest/enjoyment, perceived choice, and pressure from the IMI [35]. Within the sample, respondents had the highest central tendency (potential range of 1–7) scores for interest/enjoyment (6.71), followed by perceived choice (6.40), competence (5.60), and pressure (3.00), respectively.

Compared to the validation study mean subscale scores, respondents in this study scored low for change strategies, cons, and peer influences, and high for family influences and environmental factors from the PACE+. Within the sample, respondents had the highest central tendency scores for environmental factors (4.40), pros (4.25), self-efficacy (3.91), family influences (3.08), change strategies (2.85), peer influences (2.51), and cons (1.32) in that order.

### 3.2. Qualitative Results

#### 3.2.1. Sample Characteristics

Of the 15 interview participants, 14 were male and only one was female. The participants’ ages ranged from 7 to 16 years, with a median age of 12 years. The interview participants had a median of 36 unique visits to the gym and 12 months of attending, which is very similar to the central tendencies of the total quantitative sample.

Generally, participants shared positive feelings about Parkour, and their experiences exhibited themes of SDT, with the most common response being the freedom (autonomy) and social benefits (relatedness). Thematic interpretations of motivations expressed in interviews are grouped based on the SDT constructs of interest: autonomy, relatedness, and competence.

#### 3.2.2. Autonomy in Physical Activity

Autonomy is operationalized here as their perceived control over their choice to both participate in Parkour and their physical actions while participating in it.

##### Introduction to Parkour

The participants’ introductions to Parkour were split between dependent (introduced by a friend or family member) or independent (discovered on their own), with a greater emphasis towards independent introduction. Dependent introduction was primarily from parents who brought them to try a class, or a friend who was already taking Parkour classes. Independent introduction was facilitated by a variety of modalities, including YouTube videos, video games, TV shows, or personal desires to learn acrobatic Parkour skills. The greater emphasis towards independent introduction suggests a level of perceived autonomy in the initiation of Parkour.

##### Fluid Structure

When participants were asked what they enjoyed about Parkour, many reported preferring to take part in activities that had a less rigid structure. Having less structure was specifically stated by one third of participants as one of the unique aspects of Parkour (Figure 1). This loose structure was reported alongside feelings of creativity and fun as the major reason that students engaged in Parkour.

“I really like that it’s determined by you, there is no one telling you, oh, you have to go this way or you have to do this type of vault. You can basically like, choose it all yourself and like, kind of suit your own path. Um, so if you don’t know, you can like push yourself to you can push yourself, um, and that it’s not; it’s about running. It’s about climbing, both natural movement and basically a lot of things into one.”(Male, 13)

While many responses to this question were focused on the physical skills to be gained, concepts of Parkour being self-driven, easily accessible, non-aggressive, and unstructured emerged, once again emphasizing the self-direction perceived by the participants.

##### Personal Goals

When asked what they wanted to gain from doing Parkour, the majority reported goals that were intrinsic versus those that were extrinsic (Figure 2). Extrinsic goals included reaching higher band levels (analogous to belts in Karate) or reaching high level competitions. Intrinsic goals were primarily focused around having fun, getting stronger or more skillful, and gaining confidence. Goals intersected with lessons the participants felt they had learned through Parkour that extended beyond sports, including confidence, and hard work.

“Um, I don’t think I’m really in it to gain anything. I mean, I’d love to learn how to do cool tricks. But as long as I’m having fun and like learning new skills and hanging out with people that I would have never met, otherwise, I think I’m okay with it.” (Male 13)

This emphasis on self-improvement, fun, and other intrinsic goals despite the presence of extrinsic motivators suggests again an inward, non-competitive emergent theme from participant preferences related to Parkour.

#### 3.2.3. Physical Competence

Operationalized here as their perceived ability to accomplish their goals, both within the context of Parkour participation and outside of it.

##### Accessibility

One third of participants enjoyed that the skills they had learned in Parkour class could be done elsewhere. Variations of this idea emerged when asked what they enjoy about Parkour (Practicing what they learned at home) and what is unique to Parkour (Figure 1). Three participants focused their responses on the idea that Parkour was more accessible without the need for equipment or a specific set-up, so it was easier for them to practice. Consistent with this theme, nearly two-thirds of participants said they practice Parkour outside of the gym, primarily in places away from home.

“Um, I like it because there’s so many different, like, you could do it anywhere like with baseball, I mean yes you could do it almost anywhere. But you have to have the equipment. Parkour you have no equipment. Yes, like for baseball, you have to find a flat area. You need a bat, ball, and bases. But for Parkour, you need nothing really.”(Male, 8)

##### Unique Type of Movement

Nearly all participants mentioned Parkour movements as specifically unique, and their descriptions of it imply a sense of utility and capability. When asked generally what lessons they learned, approximately half of participants replied with a specific skill they had learned, highlighting their enjoyment of the unique skills taught in Parkour. When asked about feeling pressured in Parkour, self-imposed pressure was a major theme, with students acknowledging Parkour as challenging and them having to push themselves to overcome boundaries. Specifically, they valued skills that make them feel capable in the real world, including the ability to fall without getting hurt or the confidence to overcome both literal and figurative obstacles.

“Yeah, actually. I feel like the will I guess, I, this feels sounds weird, but the believing in yourself, that you can do things or like anything, because in Parkour, for that, you would usually use it for like jumps or, or flips or anything like that, but in the real world, believing in yourself would really come in handy.”(Male, 12)

#### 3.2.4. Social Connections and Interactions (Relatedness)

##### Family and Friend Perspectives

Described perspectives from friends and family members generally fell into three groups, admiration (“thinking it is cool”), encouragement and support, and family/peer views not being important to the participants. Most family and friends thought Parkour was “cool” or were very encouraging or supportive of them being involved, with only three interviewees suggesting that they were not influenced by others’ opinions. Despite participants generally feeling their choice to do Parkour was supported, many also said their family or friends thought it was dangerous.

“Half the time you’re kind of afraid of me like jumping off stuff. But the other times it’s they’re super enthusiastic about it. People are like, well it’s super cool. Can you do this? Can you do that? So usually are people are super supportive about it.”(Male, 17)

These responses suggest positive interactions surrounding Parkour between participants and those who do not engage in Parkour. This encouragement and support may contribute to a positive social environment.

##### Sense of Comradery

This was the strongest theme throughout all the interviews and questions. Participants discussed teamwork and social factors in every question. Notably when discussing their perspectives on competition, of those who enjoyed competition, nearly half cited their main reason being the socialization and social gathering type of competition that Parkour offers rather than the extrinsic motivators typically associated with competition (e.g., prizes). Positive team dynamics related to non-aggression and feeling collaborative with their friends were unique features to Parkour the participants found important.

The supportive atmosphere was one of the major themes of what they like about Parkour, and a few even reported enjoying pressure to do Parkour from their friends, describing it as motivating. Socializing was a driving force behind why the students had fun and felt the learning environment for Parkour was unique.

“A lot of like, the like friendships that I made, like I have a ton of friends that I met at the Parkour gym. And also, I just like getting better and learning new moves. It’s just, I don’t know. I like hanging out with my friends that I met there. And then I also just like improving with them and around them, it’s cool and I enjoy it.”(Male, 13)

## 4. Discussion

This study used a multimethod design to explore motivations for youth participation in Parkour. The findings from our analyses suggest that Parkour may address needs outlined by BPNT in alignment with SDT, and therefore Parkour may be an appropriate activity for engaging this age group in PA, warranting future research.

### 4.1. Interpretation of Findings

Generally, perspectives of Parkour and PA generally were highly favorable, with congruence across both quantitative and qualitative data sources. The Interest/enjoyment (IMI) and Pros (PACE+) subscales were among the highest mean scores from the quantitative results, and all interviewees spoke of Parkour favorably. High Self-efficacy (PACE+) and Perceived Choice (IMI) subscale scores aligned with the qualitative findings on themes of autonomy. While the qualitative participants explicitly mentioned learning skills that made them feel capable in the “real world”, the relatively lower Competence scores (IMI) did not fully represent this. Alternatively, Parkour may feel difficult due to abstract goals that are not associated with competition. When discussing their goals for Parkour, over two-thirds of the participants reported intrinsic goals related to fun and simply “getting better” (Figure 2). Relatedly, children can focus on individual improvement rather than competitive skill building. It is also possible that children who choose Parkour are less interested in team sports, or that Parkour skills may be more difficult to learn.

Despite having the lowest mean subscale score for the IMI, Pressure still scored highly relative to previous uses of the IMI [35]. This was unexpected given that Parkour does not typically include pressure to win competitions. However, the IMI’s questions related to Pressure focus on feelings of tension, but do not differentiate by source of pressure. Based on the participant interviews, it was nearly unanimous that the pressure in Parkour was not construed negatively. Participants reported feeling encouraged by their friends and coaches but felt pressured by themselves (intrinsic) to accomplish challenges. Parkour athletes often have to overcome both physical and mental barriers to solve problems and consistent with previous research confronting these challenges may cause fear/pressure [43].

From the PACE+, nearly 60% of the participants reported being in contemplative and pre-contemplative stage of change despite being actively engaged in PA. These stages of change questions ask about frequency of activity, so the relatively low scores of these participants may suggest that they are inadequately active or that children doing Parkour do not necessarily view it as PA. Nearly one-third of participants explicitly mentioned they enjoyed being active as a reason for enjoying Parkour, but fun was mentioned more frequently. Children doing Parkour may enjoy the PA that comes with the sport, but it was not the primary motivation for their participation. Future research should explore if Parkour could engage children who are not otherwise interested in participating in intentional PA or traditional sports.

Quantitative and qualitative results diverged on “Peer Support”, where the survey results showed low scores for friend support for PA (Peer Influences in the IMI). However, interview responses overwhelmingly suggested positive friend and social environments in Parkour. Participants felt their relationships with their friends and training partners were among the main reasons for enjoyment and participation. This could again support the idea that children doing Parkour do not think of it as PA, as the PACE+ explicitly asks about friend support for PA. Additionally, Parkour is not (typically) a team sport or activity, the focus on the individual could mean that relative to traditional team sports, peer influences are not as prevalent. Similarly, the PACE+ asks about being teased for doing PA, and one interview respondent noted that their friends initially thought Parkour was “weird”. While the novelty of Parkour can provide unique enjoyment (as expressed by participants) this novelty could also be alienating or more difficult to share with others. Some previous research has suggested that Novelty (reported by our participants through the framing of Parkour as “unique”) may satisfy the requirements of a BPN as well, suggesting further alignment between our findings and the theoretical constructs [44]. In the PACE+, Environmental Factors was the subscale with the highest score. Questions in this section relate to resources at home and perceptions of their PA environment, which may reflect a reframing of the way participants perceived their PA environment or viewed new places as informal PA spaces, consistent with previous research in other “lifestyle sports” such as skateboarding [45]. These survey results align with the qualitative responses, wherein participants describe practicing Parkour around their homes, neighborhoods, and in a variety of public spaces where they can find “any obstacles outside”.

The qualitative data illustrated additional themes of autonomy that were not captured by the quantitative data, as represented by the IMI under the “Choice” and ‘Pressure” subscales. Children identified the freedom and creativity of Parkour as some of their favorite aspects; while explicitly stating they felt it was their choice not only to do Parkour, but what to do while they were participating. Overall, our results strongly align with the theoretical foundation of BPNT. Qualitative findings suggest that Parkour participants highlighted themes of autonomy, relatedness, and competence in their reasons for enjoyment. Their described rationales for why they engaged in Parkour also aligned with intrinsic framings of motivation, consistent with SDT.

### 4.2. Context of Existing Research

There is a paucity of research on Parkour and its potential impacts on motivation for PA in any group. Published studies focused primarily on kinetics or kinematic research or injuries. Several studies on Parkour focused on the positive effects on cardiovascular fitness and its potential for improving athletic development in more traditional sports [46,47]. One unique randomized controlled trial found schoolchildren were more likely to participate in an organized Parkour recess versus a standard supervised recess [48]. More broadly, Parkour fits into the category of “lifestyle sports” [45,49]. Previous ethnographic research on skateboarding, as a popular example of an urban lifestyle sport, indicates that specific aspects of this more autonomous style of activity can provide rich subjective experiences that make these activities attractive [50].

Although no previous research has explicitly explored Parkour and SDT, our findings are consistent with the existing research which has indicated positive aspects of participation and health identities in a Parkour setting [51,52]. Previous qualitative research exploring the use of Parkour in a school setting resulted in five themes (enjoyment, fear, problem-solving skills, social skills, and inclusion), many of which were echoed in our interview responses [43]. One of this study’s strengths over prior research is the use of multiple methods and the guiding framework of SDT to align findings with established theory. This novelty in our approach provides initial evidence for the potential of Parkour to satisfy BPNs specifically. Even among lifestyle sports, Parkour offers a low barrier of entry given that no equipment is required and the sport can be practiced in a variety of settings. These factors may make Parkour suitable for future PA interventions.

### 4.3. Strengths and Limitations

Strengths include the use of both quantitative and qualitative methods to examine motivation in Parkour, which allowed for a deeper exploration of themes than single method analyses could accomplish. The strong basis in BPNT provided a reliable framework to explore the primary questions. All interviews were carried out by a third-party interviewer unfamiliar with any of the participants, which may have reduced response bias. Given the exploratory nature of this research, the present findings provide the basis for future research into Parkour and similar lifestyle sports through the lens of motivation.

The primary limitation of the current study was the small sample size with a large age range that was recruited from a single site using self-report measures. The 27 Parkour participants comprised approximately one-quarter of all members of the facility within the set age range at the time of data collection. Limited resources only allowed for non-random convenience sampling, which may have resulted in a biased sample. Our sample included children as young as 7, while the PACE+ survey has only been validated in children 11 and up; this discrepancy may have impacted the accuracy of responses received for that measure given limited child understanding/relevance. This age-discrepancy was partially accounted for by having parents present during the survey process. The recruitment was also limited by the timing of the COVID-19 pandemic, resulting in a smaller quantitative sample size than originally planned. Our research also included a male-dominated sample (*N* = 25; 92.6%). The small female sample may be indicative of gender representation in Parkour, with one study suggesting female representation in the sport to be around 12% [53]. Given that recruitment was limited to children who were actively enrolled at the gym, this sample may also be biased in terms of SES (capable of affording extracurricular sport activities) and also did not capture participants who had discontinued Parkour or who practice it outside of the gym setting. This small, non-representative convenience sample impacts the generalizability of these findings to females or diverse socioeconomic backgrounds that future research should aim to address especially given that motivations for activity and enjoyment of Parkour may significantly differ based on these factors.

## 5. Conclusions

This study illustrates the potential of Parkour as a PA choice for youth and provides a starting point for future research to explore motivation for participation in youth sports, particularly individual non-competitive sports. In particular, the results related to PA stage of change and change strategies from the PACE+ suggest that children who participate in Parkour may not otherwise be interested in PA. The use of qualitative methods highlighted the distinct features of Parkour that make it enjoyable.

Future research should expand on this work by exploring other individual sports focused on self-improvement, such as martial arts, surfing, and skateboarding, and by further examining intrinsic and extrinsic motivations for participating in competitive team sports. This could elucidate reasons for initiating participation in sports and, importantly, reasons that children discontinue sports as they develop through adolescence. Given the decline in activity throughout adolescence and into adulthood, a better understanding of maintaining participation in sports is especially important for designing interventions that lead to sustained increases in PA.

The results related to satisfying BPNs suggest potential for Parkour and other lifestyle sports as a PA choice for some youth. Exploring PA opportunities that offer more autonomy and address other psychological needs in youth may prove to be advantageous in their adoption and maintenance of this essential health behavior as children grow into adulthood. Promotion of PA and other health behaviors among youth requires understanding motivations and how athletic offerings satisfy those needs.

## Figures and Tables

**Figure 1 ijerph-22-01632-f001:**
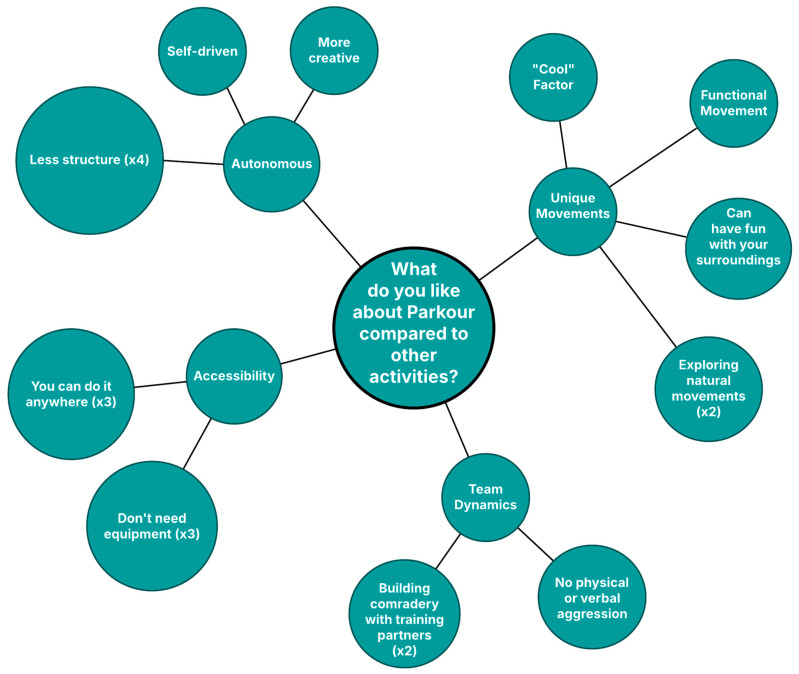
Graphical depiction of thematic responses to “What do you like about Parkour compared to other activities?”.

**Figure 2 ijerph-22-01632-f002:**
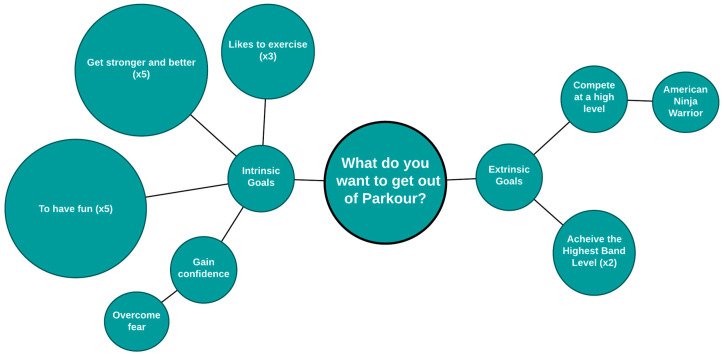
Graphical depiction of thematic responses to “What do you want to get out of Parkour?”.

**Table 1 ijerph-22-01632-t001:** Descriptive Statistics.

	*N*	Mean	SD	Median	Min	Max
Demographics						
Age	27	11.07	2.66	11.00	7.00	16.00
Months of Attendance	25	13.04	8.15	12.00	2.00	25.00
Number of Gym Visits ^a^	25	42.44	24.77	35.00	7.00	104.00
IMI ^b^						
Interest/Enjoyment	23	6.21	1.20	6.71	2.57	7.00
Competence	27	5.61	0.87	5.60	3.00	7.00
Perceived Choice	27	5.96	1.38	6.40	1.00	7.00
Pressure	27	2.99	0.93	3.00	1.40	4.60
PACE+ ^c^						
Change Strategies	27	2.80	0.80	2.85	1.37	4.07
Pros	26	3.89	1.12	4.25	1.25	5.00
Cons	25	1.44	0.48	1.32	1.00	3.02
Self-Efficacy	26	3.59	0.82	3.91	1.98	4.82
Family Influences	27	3.08	0.66	3.08	1.78	4.40
Peer Influences	27	2.57	0.59	2.51	1.31	3.68
Environmental Factors	27	4.14	0.81	4.40	2.11	5.00

^a^ Over the 6 months prior to data collection. ^b^ All IMI subscale scores range from 1–7. ^c^ All PACE+ subscale scores range from 1–5.

## Data Availability

Data can be made available upon request.

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
