# Peer review of "Parkour and Intrinsic Motivation: An Exploratory Multimethod Analysis of Self-Determination Theory in an Emerging Sport"

_ijerph, 2025, doi:10.3390/ijerph22111632_

Round 1
Reviewer 1 Report
Comments and Suggestions for Authors
Dear. author,
This paper addresses an intriguing emerging topic: parkour as a youth sport analyzed through Self-Determination Theory (SDT). Its strength lies in a multifaceted approach combining quantitative and qualitative methods. However, improvements are needed in the following areas:
- Sample Size and Representativeness: This study relies on a convenience sample (N=27, 25 males) from a single facility, which is small and male-dominated. This significantly limits generalizability. Please discuss more explicitly the impact of gender imbalance and socioeconomic bias on the results.
- Validity of measurement tools for younger children: The PACE+ survey was used with participants as young as 7 years old, although its validity is established for ages 11 and above. A more cautious explanation of this limitation is needed, and the impact on validity should be clearly discussed.
- Integration of Mixed-methods Findings: The triangulation of quantitative and qualitative data leans heavily descriptive. Stronger integration is needed, particularly to explain the discrepancy between survey findings (low peer support) and interview findings (high social connectedness).
- Novelty and Contribution: Much of the discussion reiterates known SDT applications. Clarify unique contributions: What makes parkour distinct compared to other lifestyle sports? How does this study extend theory and contribute to practical interventions?
- Conflict of Interest: While disclosed, the principal investigator's employment at a parkour facility raises potential bias concerns. Detailed safeguards (e.g., independent interviewers, parental presence) are needed to build reader trust.
- Minor Comments:
- Figures/tables (e.g., theme map) should be restructured for clarity and readability.
- Ensure consistency in terminology (“competence” vs. “competency,” ‘PA’ vs. “physical activity”
Best,
Author Response
Thank you for your review, please see the attachment.

Reviewer 2 Report
Comments and Suggestions for Authors
Article review
- The first sentence in the abstract indicates that SDT has been underutilized in studying youth sport participation. This is blatantly untrue. Researchers have used SDT in studying youth sport motivation for more than 30 years and it has a rich history. It is interesting to note that the literature review goes on to cite many instances of SDT use in study of the topic. Joan Duda is a leading researcher in this area, and a simple Google search shows over 23,000 articles on the topic. I would not begin the abstract with this sentence. I realize authors want to show the uniqueness of their project, but in this case applying the theory to parkour is enough.
- SDT has also been used to study exercise and leisure activity participation, and that may be the better application to parkour. This line of research wasn’t mentioned. I recommend reviewing studies in SDT and leisure exercise to see how those relate to the question. I’d also expand the parkour section to explain better what parkour is, the philosophy behind it and so on.
- SDT is explained pretty broadly in this article. It actually has 6 mini-theories, some of which may be applicable to the research. The authors should review the Basic Psychological Needs Theory as that seems to be the one most aligned with their research.
- Is parkour an exercise activity or competitive in this sample? Is it more like a sport or more like a leisure activity? That is an important question.
- Can the authors provide the interview protocol?
- The sample is very small for a quantitative study, and I see that no analysis other than descriptives was reported. The IMI means were compared to previous research. What previous research? What was the sample? That is very important to discuss. The results would be more definitive if there was a larger sample or a comparison group of some sort.
- I would not consider this a quantitative study, as no true analyses were conducted. It is a qualitative study with basic demographic information collected to support the interview. I’d recommend rewording how the study is described.
- I’d like to see the discussion focus on how this relates theoretically to SDT. The authors can put it in the context of one of the 6 mini-theories.
- The article is well-written and provides information on a unique and interesting activity.
Author Response

(The authors gave the same response as above.)

Round 2
Reviewer 2 Report
Comments and Suggestions for Authors
No further suggestions